# Development of Ultradeformable Liposomes with Fatty Acids for Enhanced Dermal Rosmarinic Acid Delivery

**DOI:** 10.3390/pharmaceutics13030404

**Published:** 2021-03-18

**Authors:** Thirapit Subongkot, Tanasait Ngawhirunpat, Praneet Opanasopit

**Affiliations:** 1Pharmaceutical Innovations of Natural Products Unit (PhInNat), Department of Pharmaceutical Technology, Faculty of Pharmaceutical Sciences, Burapha University, Chonburi 20131, Thailand; 2Faculty of Pharmacy, Silpakorn University, Nakhon Pathom 73000, Thailand; Ngawhirunpat_T@su.ac.th (T.N.); opanasopit_p@su.ac.th (P.O.)

**Keywords:** ultradeformable liposomes, fatty acids, rosmarinic acid, dermal drug delivery, skin penetration pathway

## Abstract

This study aimed to develop ultradeformable liposomes (ULs) with fatty acids, namely, oleic, linoleic, and linolenic acid, to improve the skin penetration of rosmarinic acid. This study also investigated the vesicle-skin interaction and skin penetration pathway of ULs with fatty acids using the co-localization technique of multifluorescently labeled particles. The prepared ULs were characterized in terms of size, surface charge, size distribution, shape, % entrapment efficiency (% EE), and % loading efficiency (% LE). The prepared ULs with fatty acids had an average particle size between 50.37 ± 0.3 and 59.82 ± 17.3 nm with a size distribution within an acceptable range and exhibited a negative surface charge. The average % EE and % LE were 9 and 24.02, respectively. The in vitro skin penetration study found that ULs with oleic acid could significantly increase the skin penetration of rosmarinic acid compared to ULs. According to confocal laser scanning microscopy observations, this study suggested that UL vesicles attach to the skin before releasing the entrapped drug to penetrate the skin. These findings suggested that ULs with oleic acid penetrated the skin via the transfollicular pathway as a major penetration pathway.

## 1. Introduction

Dermal drug delivery is the transportation of active compounds into the skin for various purposes, such as disease treatment as well as diagnosis and cosmetic application. The skin is the largest organ and consists of heterogeneous layers, including the epidermis, dermis, and subcutaneous tissue. However, the delivery of drugs into the skin is limited by the stratum corneum, which is the outermost layer of the skin. Rosmarinic acid is a polyphenolic compound that has various biological activities, such as anti-inflammatory [1], antioxidant, and antimelanogenic properties [2,3]. Due to its useful pharmacological properties, the delivery of rosmarinic acid into the skin is beneficial for skin health. Rosmarinic acid has a molecular weight of 360.3 g/mol, log partition coefficient (log P) of 1.82, and pKa of 3.57 [4].

Theoretically, compounds that have log P values between 1 and 3 with a molecular weight below 500 Daltons was considered to have good skin penetration properties [5]. However, at skin pH (pH ~ 5) [6], rosmarinic acid exhibits low solubility due to very low ionization. According to the ex vivo penetration of rosmarinic acid using full thickness human skin, rosmarinic acid was detected only in epidermis whereas rosmarinic acid inside the dermis and receiver medium was not found [7]. Thus, rosmarinic acid has low skin penetration efficiency due to its physicochemical properties.

To increase the skin penetration of drugs, various strategies have been used, such as microemulsions [8], liposomes [9], ultrasound [10], nanofibers [11], niosomes [12], and magnetophoresis [13]. Liposomes are spherical phospholipid bilayer nanoparticles that can entrap both hydrophilic and lipophilic drugs. Ultradeformable liposomes (ULs) are a type of elastic liposome generated by the addition of edge activators into liposomes. This type of liposome has been shown to increase the skin penetration of various drugs [14,15,16]. According to the in vitro skin penetration study by Subongkot et al. [14], ULs with terpenes could increase the skin penetration of sodium fluorescein more than ULs. Thus, the addition of a skin penetration enhancer for ULs is necessary for better efficacy in enhancing the skin penetration of UL.

There have been many reports of various types of skin penetration enhancers, such as surfactants, urea, terpenes, alcohols, peptides, sulfoxides, glycols, and fatty acids [17]. Many fatty acids are generally recognized as safe (GRAS) and are approved by the Food and Drug Administration (FDA) as inactive ingredients in many products. Fatty acids can be classified into two groups: Saturated fatty acids and unsaturated fatty acids. There have been reports that unsaturated fatty acids show more efficacy than saturated fatty acids for skin penetration enhancement [18,19]. There was a report using fatty acids as an enhancer for pulmonary drug delivery [20]. However, there has been no report using unsaturated fatty acids as skin penetration enhancers for ULs. Therefore, this study aimed to improve the skin penetration of rosmarinic acid using ULs with unsaturated fatty acids and to investigate the skin penetration pathway of ultradeformable liposomes. Moreover, the physicochemical properties of various rosmarinic acid loaded liposomes such as particle size, size distribution, and droplets surface charge, were investigated.

## 2. Materials and Methods

### 2.1. Materials

Rosmarinic acid, rhodamine B base, and cholesterol were purchased from Sigma-Aldrich, St. Louis, MO, USA. Phospholipids (Phospholipon 90 G) were donated by Lipoid GmbH, Ludwigshafen, Germany. Oleic acid and linoleic acid were purchased from Sigma-Aldrich, St. Louis, MO, USA. Alpha-linolenic acid (linolenic acid >70%) was purchased from Tokyo Chemical Industry Co., Ltd., Tokyo, Japan. Tween 20 was purchased from Ajax Finechem, Auckland, New Zealand. 1,2-Dihexadecanoyl-sn-glycero-3-phosphoethanolamine triethylammonium salt (NBD-PE) was purchased from Thermo Fisher Scientific, Waltham, MA, USA. 4′,6-Diamidino-2-phenylindole dihydrochloride (DAPI) was purchased from Invitrogen, Camarillo, CA, USA. All the other reagents were of analytical grade and commercially available.

### 2.2. Preparation of Rosmarinic Acid Solution and Different Rosmarinic Acid-Loaded Liposomes

The rosmarinic acid solution was prepared by weighing 6.39 mg of rosmarinic acid accurately in a volumetric flask, adding 150 µL of ethanol and stirring until dissolved. Then, water was added to adjust the volume to 5 mL. Various formulations of rosmarinic acid-loaded liposomes are shown in Table 1. There have been several methods to prepare liposomes such as the thin film method and supercritical assisted technique [21]. In this study, liposomal formulations were prepared using the conventional thin film method. The rosmarinic acid stock solution was prepared by weighing 125 mg of rosmarinic acid into a volumetric flask, and adjusting the volume to 5 mL using methanol. To prepare the phospholipid stock solution, 0.773 g of phospholipid was weighed into a glass vial, brought to 5 mL with 2:1 *v*/*v* chloroform:methanol, and stirred until completely dissolved. The cholesterol stock solution was prepared by weighing 0.0618 g of cholesterol into a glass vial, and 8 mL of 2:1 *v*/*v* chloroform:methanol was added and stirred until dissolved.

To prepare 5 mL of rosmarinic acid-loaded ULs with fatty acids, 256 µL of rosmarinic acid stock solution, 250 µL of phospholipid stock solution, and 500 µL of cholesterol stock solution were pipetted into a test tube. The solvent was removed using a soft stream of nitrogen gas until the thin film occurred throughout the test tube. The obtained thin film was kept in the desiccator overnight. Then, the thin film was hydrated with water and vortexed by a vortex mixer until it was completely dispersed. Afterwards, the liposomal dispersion was filled into a glass vial containing a mixture of Tween 20 and each fatty acid and sonicated by placing the vial in an ice bath using a 3 mm diameter probe sonicator (Vibra-Cell Processors VCX 750, SONICS & MATERIALS, INC, Newtown, CT, USA) for 30 min. The energy and amplitude of the probe sonicator were set at 10,000 Joules and 20%, respectively, throughout the sonication time.

Rosmarinic acid-loaded conventional liposomes and rosmarinic acid-loaded ULs were produced using the same method, as described above.

### 2.3. Characterization of Liposomes

#### 2.3.1. Particle Size, Polydispersity Index (PDI), and Surface Charge

The average particle size, PDI, and zeta potential of various rosmarinic acid-loaded liposomes were determined using a dynamic light scattering (DLS) particle size analyzer (Zetasizer Nano-ZS, Malvern Instrument, Worcestershire, UK) equipped with a 4 mW He–Ne laser at a scattering angle of 173°. The sample was diluted with an appropriate amount of water prior to each measurement. The measurement was performed under ambient conditions and in triplicate.

#### 2.3.2. Transmission Electron Microscopy (TEM)

TEM was used to analyze the size and shape of liposomes. Each liposomal formulation was diluted with an appropriate amount of water and sonicated in a sonicator bath. A drop of diluted liposomes was pipetted onto the formvar-coated grid, stained with 2% uranyl acetate aqueous solution, and allowed to dry at room temperature. The samples were visualized by TEM (Philips Tecnai 20; FEI/Philips Electron Optics, Eindhoven, The Netherlands) at 80 kV.

#### 2.3.3. Entrapment Efficiency (% EE) and Loading Efficiency (% LE)

The EE percentage was determined from the ratio between the amount of drug entrapped inside liposomes and total drug incorporated in liposomal formulation. Each liposomal formulation (0.5 mL) was pipetted into the sample reservoir of the centrifugal filter unit with a molecular weight cutoff of 3000 Daltons (Amicon Ultra-0.5, Merck KGaA, Darmstadt, Germany) and centrifuged at 4 °C at 10,000× *g* for 1 h. The filtrate in the retentate vial was removed, and 250 µL of water was added to the sample reservoir followed by centrifugation at 4 °C at 10,000× *g* for 40 min. Then, rosmarinic acid entrapped in liposomes was disrupted by the addition of 0.1% *w*/*v* Triton X-100 in the sample reservoir and centrifuged at 4 °C at 10,000× *g* for 10 min. The obtained filtrate was analyzed by high-performance liquid chromatography (HPLC), and the % EE was calculated from Equation (1):% EE = (*R_L_*/*R_i_*) × 100(1)

*R_L_* is the amount of rosmarinic acid entrapped in liposomes and *R_i_* is the initial amount of rosmarinic acid added to the liposomes.

The % LE was calculated using Equation (2):% LE = (*R_t_*/*L_t_*) × 100(2)

*R_t_* is the amount of rosmarinic acid entrapped in liposomal formulation and *L_t_* is the amount of phospholipid and cholesterol added to the liposomal formulation.

### 2.4. In Vitro Skin Penetration Study

#### 2.4.1. Skin Preparation

The full-thickness skin used in this study was neonatal porcine skin from piglets that died of natural causes and was provided by a local pig farm in Nakhon Pathom Province (Nakhon Pathom, Thailand). The skin was obtained from approximately 1 kg of the abdominal part of a pig. The obtained skin was stored in a refrigerator at −40 °C until use. The muscle layer that might have adhered to the skin from the collection process was removed using a surgical blade. Prior to the experiment, the skin was thawed and washed with phosphate-buffered saline (PBS) to remove any contaminants before mounting in diffusion cells.

#### 2.4.2. Skin Penetration Study

The skin penetration study of rosmarinic acid from various formulations was performed using a Franz diffusion cell apparatus. The skin was mounted between the donor and receiver compartments by turning the stratum corneum to the donor part. Each water jacketed diffusion cell was connected to a water circulating bath to maintain the temperature at 32 °C. The donor part was filled with 2 mL of the tested formulations, while the receiver compartment was filled with PBS. The donor compartment and sampling port were covered with parafilm^®^ to prevent evaporation. After the treatment for 6 h, the tested formulation in the donor part was removed, and the skin was washed with PBS three times before removal from the diffusion cell. The treated skin was wiped with tissue paper, and stratum corneum layers were removed from the skin using the tape strip method according to the method, as described by Subongkot and Sirirak [22]. The skin was fixed on an aluminum tray containing hard paraffin wax, and stratum corneum was stripped with a 24 mm wide pressure-sensitive adhesive tape (Scotch^®^ Transparent Tape 500, 3M Co., Ltd., Bangkok, Thailand) 40 times. All the stripped tapes were kept in glass vials containing 10 mL of ethanol and placed in a sonicator bath for 15 min. One milliter of the extracted ethanol was pipetted into the microcentrifuge tube and centrifuged at 11,180× *g* at 25 °C for 15 min (Thermo Scientific^TM^, model: Sorvall^TM^ Legend^TM^ XTR Centrifuge, Thermo Scientific^TM^, Waltham, MA, USA). The obtained supernatant was analyzed for the rosmarinic acid content by HPLC.

The rosmarinic acid amount in the stratum corneum was calculated from Equation (3):Drug amount in the stratum corneum (µg/cm^2^) = *Rs*/*S*(3)

*Rs* is the amount of rosmarinic acid in the stratum corneum (µg) and *S* is the skin penetration area (cm^2^).

The remaining skin from which the stratum corneum was stripped was cut into small fragments and placed into a screw cap glass vial containing 3 mL of ethanol. The vial was then placed in the sonicator bath for 15 min. Then, 1 mL of the extracted ethanol was pipetted into a microcentrifuge tube and centrifuged at 11,180× *g* at 25 °C for 15 min. The obtained supernatant was analyzed for the rosmarinic acid content by HPLC. The rosmarinic acid amount in the viable epidermis and dermis was calculated from Equation (4):Drug amount in the viable epidermis and dermis (µg/cm^2^) = *Rv*/*S*(4)

*Rv* is the amount of rosmarinic acid in the viable epidermis and dermis (µg) and *S* is the skin penetration area (cm^2^).

The enhancement ratio (ER) was calculated from Equation (5):ER = Drug amount in the viable epidermis and dermis of liposomal formulation/Drug amount in the viable epidermis and dermis of rosmarinic acid solution(5)

### 2.5. Confocal Laser Scanning Microscopy (CLSM) Study

To investigate the vesicle-skin interaction and skin penetration pathway of ultradeformable liposomes with fatty acids, the co-localization technique using multifluorescently labeled particles was utilized. Rhodamine B base (log P = 1.95) which exhibits red fluorescent was chosen as the entrapped drug due to the similarity of log P with rosmarinic acid. To identify the liposome particle, NBD-PE, a green fluorescent phospholipid surfactant was labelled. This study was performed using the formulation that provided the highest skin penetration enhancement of rosmarinic acid from Section 2.4.2.

#### 2.5.1. Preparation of Rhodamine B Base-Loaded NBD-PE-Labeled ULs

NBD-PE was accurately weighed at 8.04 mg and dissolved in 1 mL of 2:1 *v*/*v* chloroform:methanol to prepare a stock solution. To prepare rhodamine B base-loaded NBD-PE-labeled ULs, phospholipids and cholesterol were used at the same concentration as seen in Table 1 with the addition of 150 µL of NBD-PE stock solution, and rhodamine B base was used at a concentration of 0.068% *w*/*v*. Rhodamine B base-loaded NBD-PE-labeled ULs were prepared using the same process, as described in Section 2.2.

#### 2.5.2. In Vitro Skin Penetration Study

Franz diffusion cell was used to evaluate the skin penetration of multifluorescently labeled ULs according to the method, as described in Section 2.4.2. One milliliter of rhodamine B base-entrapped NBD-PE-labeled ULs was filled in the donor part without the addition of the receiving medium. After 30 min, the treated formulation in the donor part was removed, and the skin was washed with PBS to remove the excess dye prior to CLSM visualization.

#### 2.5.3. Skin Cross-Sectioning

A small piece of treated skin from Section 2.5.2 was cross-sectioned by cryomicrotome (Leica 1850, Leica Instruments GmbH, Nussloch, Germany). The frozen tissue was embedded in a tissue freezing medium and cross-sectioned into 5 µm thick sections on a positively charged slide (Bio-Optica, Milan, Italy). The slide containing sectioned tissue was stained with 10 µg/mL DAPI solution for 5 s and immersed in water to remove the excess dye. The slide was allowed to dry at room temperature and mounted with a toluene-based synthetic resin mounting medium before being covered with a coverslip.

#### 2.5.4. CLSM Visualization

An inverted confocal laser scanning microscope (Zeiss LSM 800-Airy scan, Carl Zeiss, Jena, Germany) was used to observe the treated skin from Section 2.5.2 by placing on a cover slip and turning the stratum corneum side to the 10× objective lens. In order to gain the sequential x-z plane images, the skin was visualized using a 20× objective lens. The slides containing sectioned skin from Section 2.5.3 were observed by a 10× objective lens. The microscope is equipped with four diode lasers with excitation wavelengths of 405, 488, 561, and 640 nm. The ZEN software (blue edition) was used to operate the microscope.

### 2.6. High-Performance Liquid Chromatography (HPLC) Analysis

Rosmarinic acid was quantitatively analyzed by HPLC (Agilent 1260 infinity II LC system, Agilent Technology, Santa Clara, CA, USA) using a C18 reversed-phase column (4.6 × 250 mm, 5 µm particle size, VertiSep UPS C18, Vertical, Nonthaburi, Thailand). The mobile phase was acetonitrile: 0.5% *v/v* formic acid at a ratio of 30:70 *v*/*v* using a flow rate of 1 mL/min. The sample injection volume was 20 µL. The detection wavelength was 330 nm. The standard curve of the rosmarinic acid solution was prepared in the concentration range of 1–1000 µg/mL with R^2^ = 0.9998. The limit of detection (LOD) and limit of quantification (LOQ) were 0.1 and 1 µg/mL, respectively. The accuracy of this validated method was 99.12 ± 0.27. To determine the precision, the percent relative standard deviation was 1.52.

### 2.7. Statistical Analysis

All the data were statistically analyzed using the one-way analysis of variance (ANOVA). Values of *p* < 0.05 were considered statistically significant.

## 3. Results and Discussion

### 3.1. Characterization of Liposomes

The average particle size, PDI, and zeta potential of various rosmarinic acid-loaded liposomes are shown in Table 2. The prepared liposomes had particle sizes in the range of 50–130 nm. The average particle sizes of CL and ULs were 130 and 71 nm, respectively. The average particle sizes of different ULs with fatty acids were in the range of 50–60 nm. The average particle size of CL was significantly higher than that of the ULs, ULs with oleic acid, ULs with linoleic acid, and ULs with linolenic acid. Thus, the addition of Tween 20 could reduce the particle size of liposomes. The average particle size of ULs was significantly higher than that of ULs with linoleic acid and ULs with linolenic acid. The addition of linoleic acid and linolenic acid could reduce the particle size of ULs. There was no significant difference in particle size among ULs with oleic acid, ULs with linoleic acid, and ULs with linolenic acid.

All liposomal formulations had PDIs lower than 0.4, indicating that the prepared liposomes had a size distribution within an acceptable range. The zeta potential of liposomal formulations ranged from −2.54 to −18.03 mV. Phosphatidylcholine, a major constituent of phospholipids used in this study, is a zwitterionic molecule that has positively charged functional groups and negatively charged functional groups from choline and phosphate, respectively. Phospholipids have an isoelectric point between 6 and 6.7 [23]. The neutral pH of liposomes (pH ~ 7) is higher than the isoelectric point of phospholipids. Phosphatidyl choline showed a negative charge. Rosmarinic acid has a pKa = 3.57 and ionizes to negatively charged molecules at neutral pH in liposomes. The fatty acids incorporated in ULs also ionized to negatively charged molecules from their carboxylic group. Therefore, rosmarinic acid-loaded ULs with fatty acids exhibited a negative charge. The TEM images of each liposomal formulation are shown in Figure 1. The generated liposomes were round in shape and had nanometer-scale sizes.

The effects of drug concentration added in the conventional liposome preparation on the EE and LE are shown in Figure 2. The increase in rosmarinic acid concentration from 5.55% to 16.64% of lipid weight led to a slight increase in EE. The EE reached the highest value (9%) when the rosmarinic acid concentration was 16.64% of the lipid weight (0.13% of rosmarinic acid in formulation). However, when the rosmarinic acid concentration rose above 16.64%, the EE decreased. According to the study of entrapment efficiency of hydrophobic drugs, which were teicoplanin and rifampicin in liposomes prepared by the conventional thin film method. The maximum entrapment efficiency of teicoplanin and rifampicin were 23.4% and 15.5%, respectively [24]. It is suggested that the low entrapment efficiency of the drug might result from the conventional method preparation. The % LE slightly increased and reached the highest value at 8% when the rosmarinic acid concentration increased to 16.64% lipid weight. The % LE decreased when the rosmarinic acid concentration rose above 16.64% of the lipid weight. Therefore, rosmarinic acid at a lipid weight of 16.64% was selected for the preparation of liposomes for further study.

### 3.2. In Vitro Skin Penetration Study

The amounts of rosmarinic acid that penetrated into different layers of the skin are shown in Table 3. To compare the skin penetration enhancement efficacy of various formulations, only the amounts of rosmarinic acid in the viable epidermis and dermis were considered. The amounts of rosmarinic acid in ULs, ULs with 0.5% oleic acid, ULs with 0.5% linoleic acid, and ULs with 0.5% linolenic acid were significantly greater than those in the rosmarinic acid solution. However, only ULs with 0.5% oleic acid could improve the skin penetration of rosmarinic acid more than ULs. There was no significant difference in the amounts of rosmarinic acid among ULs with 0.5% oleic acid, ULs with 0.5% linoleic acid, and ULs with 0.5% linolenic acid.

Regarding the ER, ULs, ULs with 0.5% oleic acid, ULs with 0.5% linoleic acid, and ULs with 0.5% linolenic acid could increase the skin penetration of rosmarinic acid compared to the solution by 2.26-, 9.26-, 7.37-, and 5.84-fold, respectively. Thus, ULs with 0.5% oleic acid were chosen for further study.

The stratum corneum, the rate-limiting step in percutaneous absorption, is composed of corneocytes embedded in a lipid matrix, such as ceramide, triglycerides, cholesterol, and free fatty acids [25,26]. The barrier of the stratum corneum limits skin penetration resulting from intercellular lipid lamellae [27,28] and keratin filaments inside corneocytes and corneodesmosomes that connect corneocytes between each layer of the stratum corneum [28]. According to the study by Subongkot et al. [29], ULs with d-limonene could increase the skin penetration of fluorescein sodium from degradation of corneodesmosomes, keratin filament denaturation, and intercellular lipid disruption. There is evidence showing that oleic acid could enhance the skin penetration by increasing the stratum corneum lipid fluidity and forming permeable defects inside intercellular lipids [30,31]. This study, therefore, suggested that the ULs with oleic acid could increase the skin penetration of rosmarinic acid through the mechanisms described above.

Among fatty acids, oleic acid is generally accepted as an effective skin penetration enhancer which can increase the skin penetration by penetrating to the stratum corneum and disturbing the lipid organization [31,32]. For linoleic acid, there was an evidence showing that linoleic acid had poor percutaneous absorption and must be improved using liposomes [33]. The skin penetration enhancement of rosmarinic acid from ULs with oleic acid might result from the skin penetration ability of oleic acid, in which oleic acid can penetrate the skin more than the other fatty acids.

### 3.3. CLSM Study

According to the in vitro skin penetration study, ULs with oleic acid could increase the highest amount of rosmarinic acid. ULs with oleic acid were, therefore, chosen to investigate the skin penetration pathway using the co-localization technique of multifluorescently labeled particles. Theoretically, there are three proposed skin penetration pathways, namely, the intercellular pathway, transcellular pathway, and transfollicular pathway [34,35,36].

The top view images of whole porcine skin treated with rhodamine B base-loaded NBD-PE-labeled ULs with oleic acid are shown in Figure 3. Rhodamine B base as an entrapped drug and NBD-PE, which represent UL particles, show red fluorescence and green fluorescence, as seen in Figure 3a,b, respectively. Red and green fluorescence was clearly seen at the hair follicles, indicating the localization of ULs via the transfollicular pathway. Surprisingly, Figure 3a,b shows the same deposition pattern of red and green fluorescence. These data could be used to investigate the vesicle-skin interaction of ULs. This study hypothesized the release and attachment process of drug-loaded UL vesicles based on the possibility of fluorescence color deposition. If drug-loaded ULs attach to any part of the skin before releasing the entrapped drug, the red and green fluorescence will deposit within the same region. If the drug is released before the UL vesicles attach to the skin, the red and green fluorescence will not deposit in the same region. Regarding the same color deposition in Figure 3a,b, this study suggested that drug-loaded UL vesicles might attach to any part of the skin before releasing the entrapped drug into the skin.

The serial x-z plane images of the marked area in Figure 3c are shown in Figure 4a. Figure 4b shows the intensity over the projection of the *z*-axis in Figure 4a. Red fluorescence, green fluorescence, and merged images at the 55 µm layer in Figure 4 are shown in Figure 5a–c, respectively. Figure 5a,b shows different deposition patterns of red and green fluorescence. Many green fluorescence spots appeared in Figure 5b but did not appear in Figure 5a. These green fluorescence spots indicated the existence of UL vesicles without the presence of entrapped drugs. This study suggested that UL vesicles attach to the skin surface before releasing the entrapped drugs for further penetration through the skin.

Cross-sectional images of porcine skin treated with rhodamine B base-loaded NBD-PE-labeled ULs are shown in Figure 6a–c. In Figure 6a, red fluorescence was clearly seen at the hair follicles more than the tissue region, indicating the penetration of UL vesicles via hair follicles. Figure 6b,c reveals green fluorescence, as seen in the white circles, indicating the penetration of UL vesicles into the hair follicles in which the entrapped drugs were already released to the surrounding tissues. Figure 6b,c also illustrates the penetration of UL vesicles via hair follicles as a major skin penetration pathway. These findings agreed with a previous study in which ULs with d-limonene penetrated the skin through hair follicles as a major skin penetration pathway [34]. Although skin appendages occupy approximately 0.1% of the total skin surface area [37], this study revealed that the transfollicular pathway was the major pathway for the penetration of UL vesicles. The intercluster pathway was the other pathway of absorption into pig skin, as found by Carrer et al. [38]. The intercluster pathways were the canyons that surrounded the cluster of corneocytes. This canyon depth occurred from the stratum corneum surface to the dermis. As observed by two-photon microscopy, negatively charged Rh-PE-labeled liposomes were evidently found in the intercluster pathway. It is suggested that the intercluster pathway might also be responsible for the penetration of ULs with oleic acid.

## 4. Conclusions

ULs with fatty acids were successfully prepared using oleic acid, linoleic acid or linolenic acid at a concentration of 0.5% *w*/*v*. The obtained ULs with fatty acids had a nanometer-scale size with size distribution within an acceptable range and negative surface charge. However, only ULs with oleic acid could significantly increase the skin penetration of rosmarinic acid compared to the control. The CLSM study using the co-localization technique suggested that ULs with oleic acid might attach to any part of the skin before releasing the entrapped drug to further penetrate the skin. ULs with oleic acid might penetrate the skin via the transfollicular pathway as a major pathway, while intercellular and transcellular pathways are minor penetration pathways. Penetration through the skin via hair follicles by passing the stratum corneum might also be the main mechanism for skin penetration enhancement of ULs with oleic acid.

## Figures and Tables

**Figure 1 pharmaceutics-13-00404-f001:**
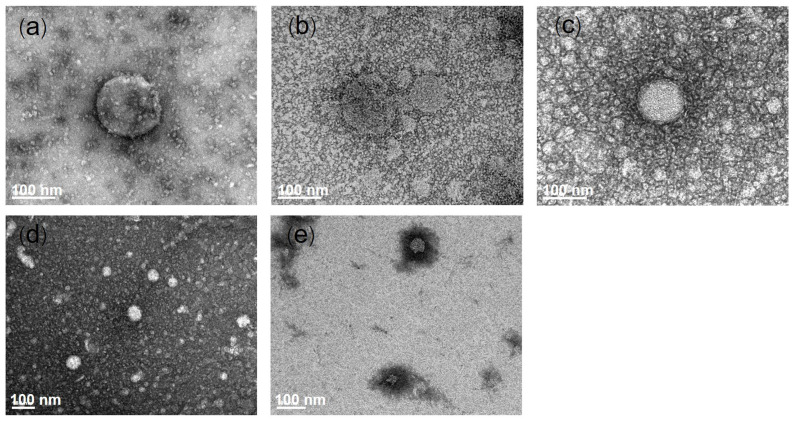
Transmission electron microscopy images of different rosmarinic acid-loaded liposomes: (**a**) Conventional liposomes (CL), (**b**) ultradeformable liposomes (ULs), (**c**) ULs with oleic acid, (**d**) ULs with linoleic acid, and (**e**) ULs with linolenic acid. The scale bar represents 100 nm.

**Figure 2 pharmaceutics-13-00404-f002:**
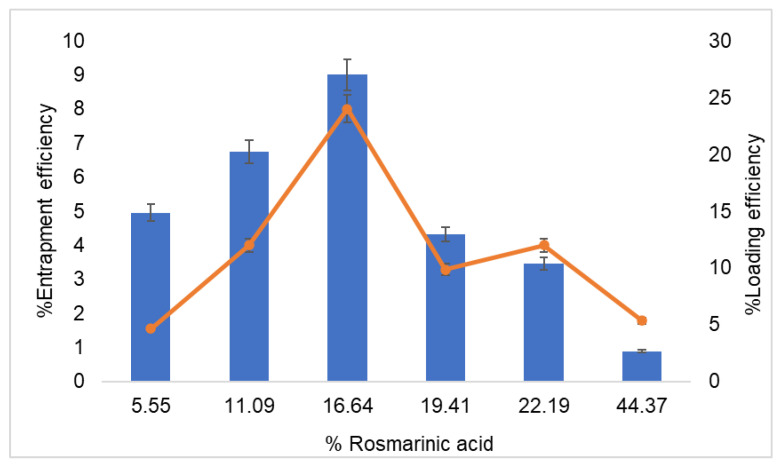
Effect of rosmarinic acid concentration (% per lipid weight) on the % entrapment efficiency (■) and loading efficiency (-). Each value represents the mean ± standard deviation (*n* = 3).

**Figure 3 pharmaceutics-13-00404-f003:**
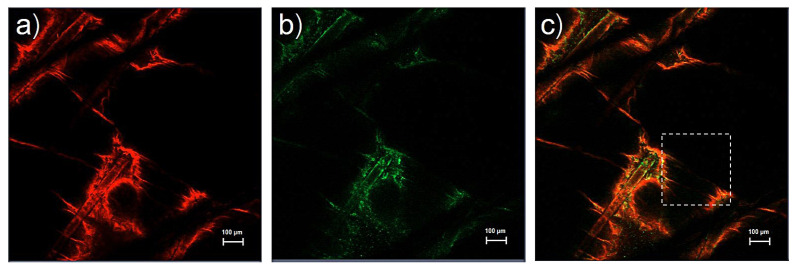
Top view images (x-y plane) of the skin treated with rhodamine B base-loaded NBD-PE-labeled ultradeformable liposomes at 30 min; (**a**) red fluorescence of rhodamine B base, (**b**) green fluorescence of NBD-PE, and (**c**) merged image of (**a**,**b**). The scale bar represents 100 μm.

**Figure 4 pharmaceutics-13-00404-f004:**
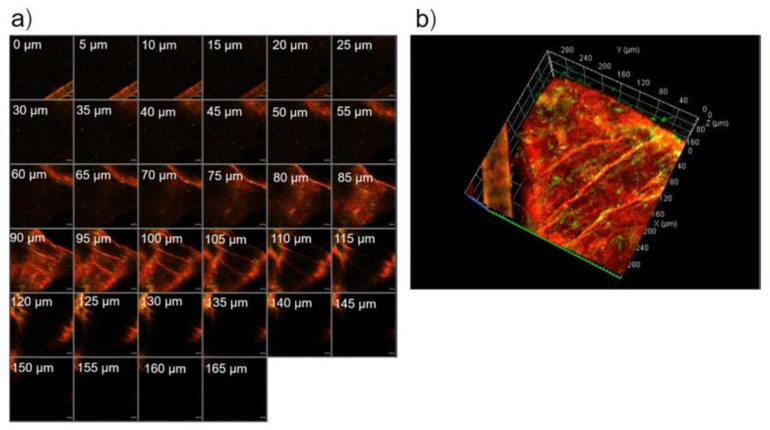
(**a**) The serial x-z plane images of the marked area in Figure 3c. The scale bar represents 20 μm, (**b**) intensity over the projection of *z*-axis in Figure 4a.

**Figure 5 pharmaceutics-13-00404-f005:**
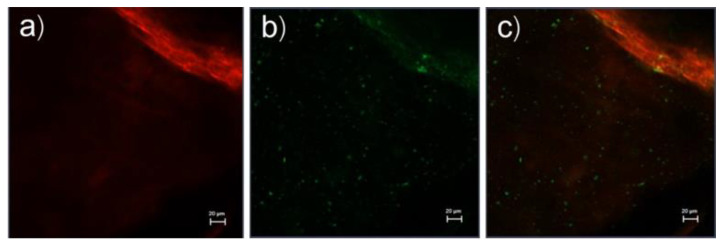
(**a**) Red fluorescence, (**b**) green fluorescence, and (**c**) merged image at the 55 µm layer in Figure 4a. The scale bar represents 20 μm.

**Figure 6 pharmaceutics-13-00404-f006:**
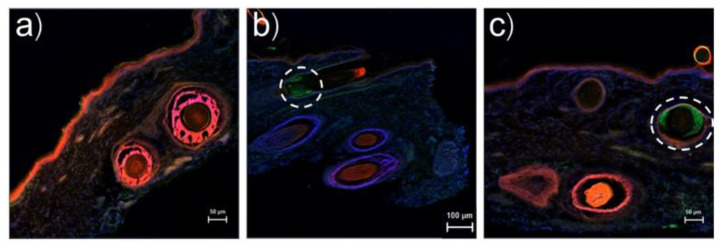
(**a**–**c**) Cross-sectional images of skin treated with rhodamine B base-loaded NBD-PE-labeled UL and stained with DAPI in different regions. The white circles indicate the presence of UL vesicles inside hair follicles in which the entrapped drugs were released.

**Table 1 pharmaceutics-13-00404-t001:** Composition of liposomal formulations.

Formulations	Rosmarinic Acid (% *w*/*v*)	Phospholipid (% *w*/*v*)	Cholesterol (% *w*/*v*)	Tween 20 (% *w*/*v*)	Fatty Acids (% *w*/*v*)	Water (mL)
Conventional liposomes (CL)	0.13	0.69	0.078	-	-	qs 100
Ultradeformable liposomes (ULs)	0.13	0.69	0.078	2	-	qs 100
ULs with oleic acid	0.13	0.69	0.078	2	0.5	qs 100
ULs with linoleic acid	0.13	0.69	0.078	2	0.5	qs 100
ULs with linolenic acid	0.13	0.69	0.078	2	0.5	qs 100

**Table 2 pharmaceutics-13-00404-t002:** The average particle size, polydispersity index (PDI), and zeta potential of various rosmarinic acid-loaded liposomes.

Formulations	Particle Size (nm)	PDI	Zeta Potential (mV)
CL	130 ± 5.1	0.40 ± 0.07	−2.89 ± 0.37
ULs	71 ± 7.5	0.28 ± 0.024	–2.54 ± 0.58
ULs with oleic acid	60 ± 17.3	0.32 ± 0.1	–18.03 ± 0.35
ULs with linoleic acid	50 ± 0.3	0.24 ± 0.008	–14.87 ± 0.86
ULs with linolenic acid	53 ± 2.3	0.27 ± 0.014	–13.20 ± 0.70

Each value represents the mean ± standard deviation (*n* = 3).

**Table 3 pharmaceutics-13-00404-t003:** Skin penetration parameters of rosmarinic acid from each formulation.

Formulations	Stratum Corneum (mcg/cm^2^)	Deeper Skin (mcg/cm^2^)	ER
Solution	2.70 ± 0.53	0.19 ± 0.03	-
ULs	0.38 ± 0.16	0.43 ± 0.22	2.26
ULs with 0.5% oleic acid	1.13 ± 0.10	1.76 ± 0.41 *	9.26
ULs with 0.5% linoleic acid	0.46 ± 0.16	1.40 ± 0.72	7.37
ULs with 0.5% linolenic acid	1.91 ± 0.64	1.11 ± 0.99	5.84

Each value represents the mean ± standard deviation (*n* = 3). Deeper skin = viable epidermis and dermis; * *p* < 0.05 compared to the ULs.

## Data Availability

The data presented in this study is contained within this article.

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
