# Peer review of "Development of Ultradeformable Liposomes with Fatty Acids for Enhanced Dermal Rosmarinic Acid Delivery"

_pharmaceutics, 2021, doi:10.3390/pharmaceutics13030404_

Round 1

Reviewer 1 Report

This paper documents the formulation of liposomes with rosemarinic acid, and the use of fatty acids in the formulation to promote skin absorption. The results are interesting and merit publication in the present form. I have no specific suggestions for further work or improvement.

A couple of suggestions for future work. The authors have carefully documented rosemarinic acid incorporation into liposomes, which seems to be limited to 12-16%. This might be improved by using a cationic lipid such as DOTAP, which might ion pair with the rosemarinic acid. Alternately, aqueous capture of the sodium salt of rosemarinic acid might be more efficient with appropriate choice of liposome preparation method, e.g. the DRV method of Kirby and Gregoriadis.

Author Response

Thank you very much for your suggestion. 

This paper documents the formulation of liposomes with rosemarinic acid, and the use of fatty acids in the formulation to promote skin absorption. The results are interesting and merit publication in the present form. I have no specific suggestions for further work or improvement.

A couple of suggestions for future work. The authors have carefully documented rosemarinic acid incorporation into liposomes, which seems to be limited to 12-16%. This might be improved by using a cationic lipid such as DOTAP, which might ion pair with the rosemarinic acid. Alternately, aqueous capture of the sodium salt of rosemarinic acid might be more efficient with appropriate choice of liposome preparation method, e.g. the DRV method of Kirby and Gregoriadis.

Thank you very much for your suggestion. We will try it for further study.

Reviewer 2 Report

The authors proposed a quite interesting paper about the production of liposomes for the entrapment of rosmarinic acid for skin penetration applications. I think that the paper can be published after major revisions.

Abstract. Line 12. “oleic acid, linoleic acid and linolenic acid”. Linoleic acid appears twice.

Line 15. “The prepared ULs were characterized for size, surface charge, 15 size distribution, shape,” I would say “… in terms of size, …”

Lien 17. “average particle size between 50.37 and 59.82 17 nm”. How could you be so precise to say that liposomes population have a mean size of 50.37 nm? 0.37 nm is just a calculation coming from the average, determined from the instrument. I would approximate to unity and, maybe, add the standard deviation.

Introduction. Line 41. There is a problem of format here.

Line 45. “Toincrease”. Change with spaces “to increase”

Line 62. “There has been no report using unsaturated fatty acids as skin penetration enhancers for ULs.”. define other applications for unsaturated fatty acids, such as:

Ghadiri, M., Canney, F., Pacciana, C., Colombo, G., Young, P. M., & Traini, D. (2018). The use of fatty acids as absorption enhancer for pulmonary drug delivery. International journal of pharmaceutics541(1-2), 93-100.

The state of the art lacks of information about the production methods for liposomes. The conventional method of thin layer hydration is the most used; however, it is not the only one. An example of word regarding the non-conventional production of liposomes for the entrapment of linalool is:

Trucillo, P., Campardelli, R., & Reverchon, E. (2019). Antioxidant loaded emulsions entrapped in liposomes produced using a supercritical assisted technique. The Journal of Supercritical Fluids154, 104626.

Regarding skin effects of linoleic acid into liposomes is also reported here:

Shigeta, Y., Imanaka, H., Ando, H., Ryu, A., Oku, N., Baba, N., & Makino, T. (2004). Skin whitening effect of linoleic acid is enhanced by liposomal formulations. Biological and Pharmaceutical Bulletin27(4), 591-594.

Line 120. Maybe % could be substituted by percentage.

I suggest adding an abbreviation list according to this journal guidelines.

Data of table 2. Same observations raised in the introduction section.

Line 251. How did you measure the pH of these liposomes? Did you adjust pH or just had a neutral value after production?

Table 2. How could be the PDI from 0.24 to 0.39 and the standard deviations be as small as 0.3 nm, 5 nm, 7 nm and so on? From PDI data, it is difficult to say that the samples are monodispersed. Instead, from SD values, they seem to be. Do you have any Particle Size Distribution diagrams for these samples?

TEM images. Some of these images seem to have a corona around them, especially the last one. Is this linked to particular phenomenon that occurs in the aqueous external bulk?

Figure 2. the entrapment efficiency is particularly low. This is probably due to the use of a conventional method that is generally particularly famous for low entrapment efficiencies. Do you have analogous situations in literature?

Could you define ER in Table 3. Does this have unit of measure or it is adimensional?

Line 369. “0.5%” means on mass basis?

Author Response

Thank you very much for your suggestion. 

The authors proposed a quite interesting paper about the production of liposomes for the entrapment of rosmarinic acid for skin penetration applications. I think that the paper can be published after major revisions.

  1. Line 12. “oleic acid, linoleic acid and linolenic acid”. Linoleic acid appears twice.

It did not appear twice. This study investigates the effect of 3 fatty acids which were oleic acid, linoleic acid and linolenic acid.

  1. Line 15. “The prepared ULs were characterized for size, surface charge, 15 size distribution, shape,” I would say “… in terms of size, …”

Thank you very much. This was edited as seen in the abstract part at line 6 of page 2.

  1. Line 17. “average particle size between 50.37 and 59.82 17 nm”. How could you be so precise to say that liposomes population have a mean size of 50.37 nm? 0.37 nm is just a calculation coming from the average, determined from the instrument. I would approximate to unity and, maybe, add the standard deviation.

The standard deviation was added as seen in line 9 of page 2.

  1. Line 41. There is a problem of format here.

Thank you, it might be format error.

  1. Line 45. “Toincrease”. Change with spaces “to increase”

It was edited as seen in line 19 of page 3.

  1. Line 62. “There has been no report using unsaturated fatty acids as skin penetration enhancers for ULs.”. define other applications for unsaturated fatty acids, such as:

Ghadiri, M., Canney, F., Pacciana, C., Colombo, G., Young, P. M., & Traini, D. (2018). The use of fatty acids as absorption enhancer for pulmonary drug delivery. International journal of pharmaceutics, 541(1-2), 93-100.

Thank you for your suggestion, this reference was added as seen in line 12-13 of page 4.

  1. The state of the art lacks of information about the production methods for liposomes. The conventional method of thin layer hydration is the most used; however, it is not the only one. An example of word regarding the non-conventional production of liposomes for the entrapment of linalool is:

Trucillo, P., Campardelli, R., & Reverchon, E. (2019). Antioxidant loaded emulsions entrapped in liposomes produced using a supercritical assisted technique. The Journal of Supercritical Fluids, 154, 104626.

This reference was added as seen in line 18-21 of page 5.

  1. Regarding skin effects of linoleic acid into liposomes is also reported here:

Shigeta, Y., Imanaka, H., Ando, H., Ryu, A., Oku, N., Baba, N., & Makino, T. (2004). Skin whitening effect of linoleic acid is enhanced by liposomal formulations. Biological and Pharmaceutical Bulletin, 27(4), 591-594.

This reference was added and discussed as seen in line 22-23 of page 15 and line 1-5 of page 16.

  1. Line 120. Maybe % could be substituted by percentage.

It was replaced by percentage as seen in line 12 of page 7.

  1. I suggest adding an abbreviation list according to this journal guidelines.

The abbreviation list was added as seen in page 1.

  1. Data of table 2. Same observations raised in the introduction section.

We added it in the introduction part as seen in line 17-19 of page 4.

  1. Line 251. How did you measure the pH of these liposomes? Did you adjust pH or just had a neutral value after production?

 The pH of liposomal formulation was measured in our preliminary study. We did not adjust pH.

  1. Table 2. How could be the PDI from 0.24 to 0.39 and the standard deviations be as small as 0.3 nm, 5 nm, 7 nm and so on? From PDI data, it is difficult to say that the samples are monodispersed. Instead, from SD values, they seem to be. Do you have any Particle Size Distribution diagrams for these samples?

We did not have the Particle Size Distribution diagrams. According to the PDI data, the value lower than 0.4is typically accepted as narrow size distribution.

  1. TEM images. Some of these images seem to have a corona around them, especially the last one. Is this linked to particular phenomenon that occurs in the aqueous external bulk?

For TEM analysis, this method is negative staining. We think that it might be the residual of uranyl acetate.

  1. Figure 2. the entrapment efficiency is particularly low. This is probably due to the use of a conventional method that is generally particularly famous for low entrapment efficiencies. Do you have analogous situations in literature?

Thank you for your suggestion, we found a literature which provide low entrapment efficiency of hydrophobic drugs. This was discussed as seen in line 9-14 of page 14.

  1. Could you define ER in Table 3. Does this have unit of measure or it is adimensional?

Enhancement ratio is the comparison of the skin penetration ability of tested formulations to control. It does not have unit.

  1. Line 369. “0.5%” means on mass basis?

It was 0.5 % w/v. It was added as seen in line 13 of page 18.

Reviewer 3 Report

The manuscript is carefully prepared. I have an impression the the work was carefully done.

I have only minor issues. It is incorrect to write Dalton. It should be either Da or daltons. It is preferable to write  g/mol.

Linolenic acid. Should it be alpha-linolenic acid?

What is a difference in the effects of similar-structured linoleic and alpha-linolenic acids?

Table 1, line 90. What is "qs"? What is "%w/v" ? It is recommended to write not ml but mL and similar liter-containing abbreviations.

Author Response

Thank you very much for your suggestion. 

  1. The manuscript is carefully prepared. I have an impression the work was carefully done.

     Thank you very much.

  1. I have only minor issues. It is incorrect to write Dalton. It should be either Da or daltons. It is preferable to write g/mol.

     It was edited as g/mol as seen in line 10 of page 3.

  1. Linolenic acid. Should it be alpha-linolenic acid?

    This was edited as seen in line 6 of page 5.

  1. What is a difference in the effects of similar-structured linoleic and alpha-linolenic acids?

     In this study linoleic acid and alpha-linolenic acid did not provide different effect on skin penetration.

  1. Table 1, line 90. What is "qs"? What is "%w/v" ? It is recommended to write not ml but mL and similar liter-containing abbreviations.

    Qs is a latin term means sufficient quantity to make. The abbreviation of mL and µL was edited through all the manuscript.

Round 2

Reviewer 2 Report

Authors provided a new version of the manuscript that has much improved. I only have a few issues, after that the paper can be surely published:

1) issue 1 was: Line 12. “oleic acid, linoleic acid and linolenic acid”. Linoleic acid appears twice.

Answer 1 was: It did not appear twice. This study investigates the effect of 3 fatty acids which were oleic acid, linoleic acid and linolenic acid.

Revised version 2 was: I am sorry, my mistake during reading of similar words.

2) Issue 2 was: Line 17. “average particle size between 50.37 and 59.82 17 nm”. How could you be so precise to say that liposomes population have a mean size of 50.37 nm? 0.37 nm is just a calculation coming from the average, determined from the instrument. I would approximate to unity and, maybe, add the standard deviation.

Answer 2 was: The standard deviation was added as seen in line 9 of page 2.

Revised version 2: Perhaps maybe authors did not catch my opinion. I said that it is not possible to have two decimal digits for this measurements. How can you assess that liposome size is 50.37 nm, and not 50 nm, or 50.4 nm? The 2 decimal digits are coming out from instrument average calculation, and is a nonsense. I suggest to approximate to unity: Such as  50±0.3 and 60±17.3 nm

I found this same problem in Table 2 values for mean size, SD and PDI.

3) issue number 3 was: Table 2. How could be the PDI from 0.24 to 0.39 and the standard deviations be as small as 0.3 nm, 5 nm, 7 nm and so on? From PDI data, it is difficult to say that the samples are monodispersed. Instead, from SD values, they seem to be. Do you have any Particle Size Distribution diagrams for these samples?

Asnwer was: We did not have the Particle Size Distribution diagrams. According to the PDI data, the value lower than 0.4is typically accepted as narrow size distribution.

Answer to revised version 2: It could be accepted, but 0.4 cannot be considered as monodisperse population. It could be only under 0.2 or even 0.15

Author Response

2) Issue 2 was: Line 17. “average particle size between 50.37 and 59.82 17 nm”. How could you be so precise to say that liposomes population have a mean size of 50.37 nm? 0.37 nm is just a calculation coming from the average, determined from the instrument. I would approximate to unity and, maybe, add the standard deviation.

Answer 2 was: The standard deviation was added as seen in line 9 of page 2.

Revised version 2: Perhaps maybe authors did not catch my opinion. I said that it is not possible to have two decimal digits for this measurements. How can you assess that liposome size is 50.37 nm, and not 50 nm, or 50.4 nm? The 2 decimal digits are coming out from instrument average calculation, and is a nonsense. I suggest to approximate to unity: Such as 50±0.3 and 60±17.3 nm

I found this same problem in Table 2 values for mean size, SD and PDI.

 Thank you for your suggestion, we have edited it as seen in line 9 of page 2, line 5-7 of page 13 and Table 2.

3) issue number 3 was: Table 2. How could be the PDI from 0.24 to 0.39 and the standard deviations be as small as 0.3 nm, 5 nm, 7 nm and so on? From PDI data, it is difficult to say that the samples are monodispersed. Instead, from SD values, they seem to be. Do you have any Particle Size Distribution diagrams for these samples?

Asnwer was: We did not have the Particle Size Distribution diagrams. According to the PDI data, the value lower than 0.4is typically accepted as narrow size distribution.

Answer to revised version 2: It could be accepted, but 0.4 cannot be considered as monodisperse population. It could be only under 0.2 or even 0.15

Thank you for your suggestion, we have edited it to “the prepared liposomes had size distribution within acceptable range” as seen in line 9 of page 2 and line 15 of page 13.
